# Verification of Neural Control Barrier Functions with Symbolic Derivative Bounds Propagation

**Hanjiang Hu    Yujie Yang    Tianhao Wei    Changliu Liu**
The Robotics Institute, Carnegie Mellon University
hanjianghu@cmu.edu, cliu6@andrew.cmu.edu

**Abstract:** Control barrier functions (CBFs) are important in safety-critical systems and robot control applications. Neural networks have been used to parameterize and synthesize CBFs with bounded control input for complex systems. However, it is still challenging to verify pre-trained neural networks CBFs (neural CBFs) in an efficient symbolic manner. To this end, we propose a new efficient verification framework for ReLU-based neural CBFs through symbolic derivative bound propagation by combining the linearly bounded nonlinear dynamic system and the gradient bounds of neural CBFs. Specifically, with Heaviside step function form for derivatives of activation functions, we show that the symbolic bounds can be propagated through the inner product of neural CBF Jacobian and nonlinear system dynamics. Through extensive experiments on different robot dynamics, our results outperform the interval arithmetic based baselines in verified rate and verification time along the CBF boundary, validating the effectiveness and efficiency of the proposed method with different model complexity. The code can be found at https://github.com/intelligent-control-lab/verify-neural-CBF.

**Keywords:** Learning for control, control barrier function, formal verification

## 1  Introduction

Safe control is of great importance in online decision making for robot learning through filtering out unsafe explorative actions [1, 2, 3, 4], guaranteeing safety during sim-to-real transfer in a hierarchical manner [5]. As an effective tool of safe control, control barrier functions (CBFs) have been studied for years on both verification and synthesis [6, 7, 8, 9, 10]. A valid CBF guarantees safety by ensuring the function values non-positive for any states along the safe trajectory, implicitly enforcing the non-trivial *forward invariance* that feasible control inputs always exist to maintain the following non-positive energy values once the state is safe. To ensure *forward invariance*, polynomial-based CBFs have been proposed based on hand-crafted parametric functions [11, 12, 13], which can be verified through algebraic geometry techniques like sum-of-squares (SOS) optimization [14, 15, 16]. However, polynomial-based CBFs cannot encode complicated safety constraints [17], and the generic parameterization of non-conservative safe control for various nonlinear dynamics and nonconvex safety specifications is needed.

Neural network parameterized CBFs (neural CBFs) have shown promising results due to their powerful expressiveness in modeling complex dynamics with bounded control inputs [18, 19, 20, 21, 22]. But it is challenging to guarantee that the learned neural CBF is valid because of its poor mathematical interpretability. One way to ensure the safety/stability of the dynamic system is to learn neural barrier/Lyapunov certificates [23, 24, 25, 26, 27, 28, 29], but it may be too conservative and cause false negatives if the formal verification only relies on a specific control policy [17, 30]. Recent works to directly verify the *forward invariance* of learned neural CBFs are based on SMT-based counterexample falsification [31, 32], mixed integer programming [33], Lipschitz neural networks [34], and CROWN-based linear bound propagation [35, 36, 37]. Although linear bound based verifi-

8th Conference on Robot Learning (CoRL 2024), Munich, Germany.

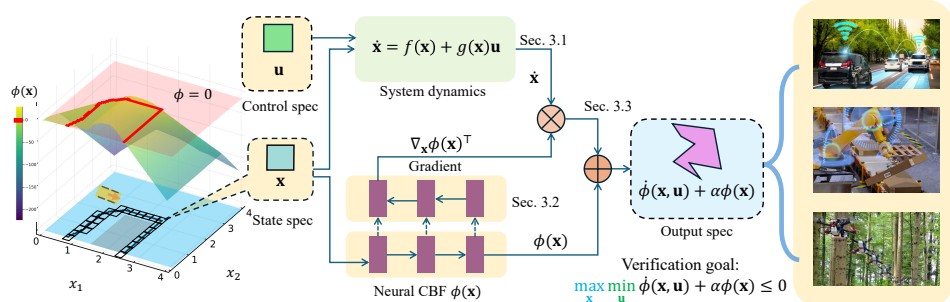

Figure 1: Overview of the verification pipeline with symbolic bound propagation, from input specifications of boundary state and control to the output specification of the CBF condition.

cation methods have shown promising scalability to larger neural networks [38, 39, 40], the bounds do not have linear symbolic format when calculating the inner product of neural networks Jacobian and the nonlinear system dynamics [36], which may be too loose and not efficient due to interval arithmetic.

Therefore, we introduce an efficient incomplete verification framework for ReLU-based neural CBFs using the fully linear symbolic bound propagation w.r.t state by combining the gradient of CBFs and neural dynamics, as shown in Fig. 1. By leveraging the fact that the derivative of ReLU activation function is the Heaviside step function, we show that linear symbolic bounds can be propagated through the inner product. Extensive experiments on various robot dynamic systems show that our method surpasses current state-of-the-art verification baselines in terms of verified rate and verification time along the CBF boundary. In summary, our contributions are listed below.

- We propose a novel neural CBF verification framework using derivative bound propagation to efficiently establish symbolic bounds for verifying forward invariance.

- We simplify the problem of propagating linear bounds through the inner product between neural CBF Jacobian and system dynamics.

- Extensive experiments validate that our proposed method achieves a 20% higher verified rate with less total verification time to verify the boundary of neural CBFs compared to the baselines.

## 2 Problem Formulation

### 2.1 Safe Control with Neural CBF

Given a control-affine system $\dot{\mathbf{x}} = f(\mathbf{x}) + g(\mathbf{x})\mathbf{u}$ with state $\mathbf{x} \in \mathcal{X} \subset \mathbb{R}^n$ and bounded control input $\mathbf{u} \in \mathcal{U} \subset \mathbb{R}^m$ and user-specified safe set $\mathcal{X}_0 \subseteq \mathcal{X}$, neural control barrier function (neural CBFs) are defined as continuous and piecewise smooth functions $\phi : \mathbb{R}^n \to \mathbb{R}$ with neural network parameterization. Generally, the neural CBF consists of $L$ feedforward layers $y_i : \mathbb{R}^{n_{i-1}} \to \mathbb{R}^{n_i}, i = 1, 2, \ldots, L$, associated with $L - 1$ ReLU activation functions $\sigma_i : \mathbb{R}^{n_i} \to \mathbb{R}^{n_i}, i = 1, 2, \ldots, L - 1$ after each linear feedforward layer. Therefore, the neural CBF is formulated as the compositional function $\phi(\mathbf{x}) = y_L \circ \sigma_{L-1} \circ y_{L-1} \circ \sigma_{L-2} \circ y_{L-2} \cdots \circ \sigma_1 \circ y_1(\mathbf{x})$, where $y_i : \mathbb{R}^{n_{i-1}} \to \mathbb{R}^{n_i}$ and $n_0 = n, n_L = 1$. Furthermore, each feedforward layer $y_i$ is parameterized with weight $\mathbf{W}_i \in \mathbb{R}^{n_i \times n_{i-1}}$ and bias $\mathbf{b}_i \in \mathbb{R}^{n_i}$ as $\hat{\mathbf{z}}_i = y_i(\mathbf{z}_{i-1}) = \mathbf{W}_i \mathbf{z}_{i-1} + \mathbf{b}_i$, where $\hat{\mathbf{z}}_{i-1} \in \mathbb{R}^{n_i}$ is the pre-activation output of $y_i$ and $\mathbf{z}_{i-1} \in \mathbb{R}^{n_{i-1}}$ is the output of last activation layer $\mathbf{z}_{i-1} = \sigma_{i-1}(\hat{\mathbf{z}}_{i-1})$. A neural CBF parameterized with neural network $\theta$ is valid if the sublevel set $\mathcal{X}_\phi := \{\mathbf{x} \in \mathcal{X} \mid \phi_\theta(\mathbf{x}) \leq 0\} \subseteq \mathcal{X}_0$ satisfies forward invariance, as defined below. In the absence of ambiguity, we omit the subscript $\theta$ and directly use the notation $\phi$ to represent neural CBF.

**Definition 1.** *A set $\mathcal{X}_\phi$ is forward invariant if for state $\mathbf{x}(0) \in \mathcal{X}_\phi$ at time $t = 0$, at any following time $t > 0$, there always exists control input $\mathbf{u}(t) \in \mathcal{U}$ such that $\mathbf{x}(t) \in \mathcal{X}_\phi$ according to $\dot{\mathbf{x}} = f(\mathbf{x}) + g(\mathbf{x})\mathbf{u}$.*

By Nagumo theorem [41], the forward invariance holds if the following boundary condition holds,

**Theorem 1** (from [11, 6]). *Given neural CBF $\phi$, if the following boundary condition over boundary set $\partial\mathcal{X}_\phi := \{\mathbf{x} \in \mathcal{X} \mid \phi(\mathbf{x}) = 0\}$ holds for the sublevel set $\mathcal{X}_\phi := \{\mathbf{x} \in \mathcal{X} \mid \phi(\mathbf{x}) \leq 0\}$,*

$$\forall \mathbf{x} \in \partial\mathcal{X}_\phi := \{\mathbf{x} \in \mathcal{X} \mid \phi(\mathbf{x}) = 0\}, \exists \mathbf{u} \in \mathcal{U}, \ s.t. \ \dot{\phi}(\mathbf{x}, \mathbf{u}) = \nabla_\mathbf{x}\phi^\top(f(\mathbf{x}) + g(\mathbf{x})\mathbf{u}) \leq 0, \quad (1)$$

*then the sublevel set $\mathcal{X}_\phi = \{\mathbf{x} \in \mathcal{X} \mid \phi(\mathbf{x}) \leq 0\}$ is forward invariant.*

Due to the universal representability of neural networks, neural CBFs have potential superiority in modeling complex forward invariant sets, showing that it is of great significance to verify that learned neural CBFs are valid.

## 2.2 Goal of Verifying Nerual CBF

Given a pre-trained neural CBF $\phi$, our goal is to formally verify that it is valid in two aspects. One is to ensure that the sublevel set $\mathcal{X}_\phi := \{\mathbf{x} \in \mathcal{X} \mid \phi(\mathbf{x}) \leq 0\}$ is a valid subset of the user-specified safe set $\mathcal{X}_0$, and the other is to guarantee the forward invariance in Definition 1. Since the neural CBF can be bounded above by $\phi_C = \phi + C, C > 0$ to shrink the forward invariance set into the user-specified safe set, without loss of generality, we reasonably assume that $\mathcal{X}_\phi := \{\mathbf{x} \in \mathcal{X} \mid \phi(\mathbf{x}) \leq 0\} \subseteq \mathcal{X}_0$ holds, and mainly focus on the verification of boundary condition (1). More generally, for better local attractiveness of the boundary in the discrete-time CBF-based safe control when sequential states cross the boundary, we incorporate non-negative constant $\alpha \in \mathbb{R}$ into Equation (1) and adopt the following equivalent condition in a more general form with $\alpha \geq 0$ [11, 17],

$$\forall \mathbf{x} \in \partial\mathcal{X}_\phi, \exists \mathbf{u} \in \mathcal{U}, \ s.t. \ \dot{\phi}(\mathbf{x}, \mathbf{u}) + \alpha\phi(\mathbf{x}) = \nabla_\mathbf{x}\phi^\top(f(\mathbf{x}) + g(\mathbf{x})\mathbf{u}) + \alpha\phi(\mathbf{x}) \leq 0. \quad (2)$$

Rewriting the condition using minimax, the final verification goal in this work is to prove

$$\max_{\mathbf{x} \in \partial\mathcal{X}_\phi} \min_{\mathbf{u} \in \mathcal{U}} \dot{\phi}(\mathbf{x}, \mathbf{u}) + \alpha\phi(\mathbf{x}) = \max_{\mathbf{x} \in \partial\mathcal{X}_\phi} \min_{\mathbf{u} \in \mathcal{U}} \nabla_\mathbf{x}\phi^\top f(\mathbf{x}) + \nabla_\mathbf{x}\phi^\top g(\mathbf{x})\mathbf{u} + \alpha\phi(\mathbf{x}) \leq 0. \quad (3)$$

# 3 Symbolic Derivative Bounds Propagation

In this section, we present a method using symbolic bound propagation to verify Equation (3). Since it is hard to precisely characterize the boundary $\mathbf{x} \in \partial\mathcal{X}_\phi$ in Equation (3), we first relax it as the union set of $K$ hyper-rectangles to over-approximate all the roots of $\phi(\mathbf{x}) = 0$ following [18, 17], $\partial\mathcal{X}_\phi \subset \Delta\mathcal{X}_\phi := \cup_{k=1}^K \Delta\mathcal{X}^{(k)}$, where $\Delta\mathcal{X}^{(k)} := [\underline{\mathbf{x}}^{(k)}, \overline{\mathbf{x}}^{(k)}] = \{\mathbf{x} \mid \underline{\mathbf{x}}^{(k)} \leq \mathbf{x} \leq \overline{\mathbf{x}}^{(k)}\}$. We then mainly focus on the verification of $\max_{\mathbf{x} \in \Delta\mathcal{X}^{(k)}} \min_{\mathbf{u} \in \mathcal{U}} \dot{\phi}(\mathbf{x}, \mathbf{u}) + \alpha\phi(\mathbf{x})$ as an exchangeable goal when referring Equation (3) with hyper-rectangle bounded input $\mathbf{u} \in \mathcal{U} := [\underline{\mathbf{u}}, \overline{\mathbf{u}}] = \{\mathbf{u} \in \mathbb{R}^m \mid \underline{\mathbf{u}} \leq \mathbf{u} \leq \overline{\mathbf{u}}\}$ and each hyper-rectangle bounded state $\Delta\mathcal{X}^{(k)}$. For simplicity, we omit the super-script $(k)$ in the following context. Our method first finds a piece-wise linear symbolic upper bound for the minimax expression in Equation (3), then resorts to out-of-the-shelf neural network verification algorithms to generate the proof. We use three steps to obtain the symbolic upper bound as shown in Figure 1. In Section 3.1, we first find the linear symbolic bounds for dynamic propagation in the inner minimization of Equation (3). We then characterize the linear bounds for $\nabla_\mathbf{x}\phi$ using gradient reachability analysis in Section 3.2. Lastly, we introduce a tight method to compute symbolic propagation through the inner product between the linear symbolic bounds for dynamic propagation and the linear bounds for $\nabla_\mathbf{x}\phi$ in Section 3.3.

## 3.1 Linear Symbolic Bounds for Dynamics Propagation with Optimal Control Inputs

To simplify the verification goal in Equation (3) by solving the inner minimization of control input $\mathbf{u}$ over hyper-rectangle $\mathcal{U}$, instead of traversing the vertices in literature [18, 36], we have the following proposition to efficiently give the equivalent expression of $\min_{\mathbf{u} \in \mathcal{U}} \dot{\phi}(\mathbf{x}, \mathbf{u})$ with the optimal control input $\mathbf{u}_v$. Proof can be found in Appendix A.1.

**Proposition 1.** *When $\mathbf{u}$ is within a hyper-rectangle $\mathcal{U} = [\underline{\mathbf{u}}, \overline{\mathbf{u}}] = \{\mathbf{u} \in \mathbb{R}^m \mid \underline{\mathbf{u}} \leq \mathbf{u} \leq \overline{\mathbf{u}}\}$, given $\mathbf{x} \in \mathcal{X}$, the minimum value of $\dot{\phi}(\mathbf{x}, \mathbf{u})$ over $\mathbf{u} \in \mathcal{U}$ can be found explicitly as,*

$$\min_{\mathbf{u} \in \mathcal{U}} \dot{\phi}(\mathbf{x}, \mathbf{u}) = \nabla_\mathbf{x}\phi^\top f(\mathbf{x}) + [\nabla_\mathbf{x}\phi^\top g(\mathbf{x})]_+\underline{\mathbf{u}} + [\nabla_\mathbf{x}\phi^\top g(\mathbf{x})]_-\overline{\mathbf{u}} = \dot{\phi}(\mathbf{x}, \mathbf{u}_v(\mathbf{x})), \quad (4)$$

where $[*]_+ = \max\{0, *\}, [*]_- = \min\{0, *\}$ *and the optimal control input* $\mathbf{u}_v(\mathbf{x}) = \arg\min_{\mathbf{u} \in V(\mathcal{U})} \dot{\phi}(\mathbf{x}, \mathbf{u})$ *lies among the vertices* $V(\mathcal{U})$ *of hyper-rectangle* $\mathcal{U}$ *given state* $\mathbf{x}$.

Suppose the control-affine system $h(\mathbf{x}, \mathbf{u}) = f(\mathbf{x}) + g(\mathbf{x})\mathbf{u}$ is analytical and its second-order derivative exists w.r.t. $\mathbf{x}$, with the optimal control input $\mathbf{u}_v \in [\underline{\mathbf{u}}, \overline{\mathbf{u}}]$ from Equation (4), the linear lower and upper bounds w.r.t $\mathbf{x}$ can be found based on 1-order Taylor models [42, 43, 44] as follows,

$$\underline{h}(\mathbf{x}, \mathbf{u}_v) = \underline{\mathbf{W}}_v \mathbf{x} + \underline{\mathbf{b}}_v \leq h(\mathbf{x}, \mathbf{u}_v) = f(\mathbf{x}) + g(\mathbf{x})\mathbf{u}_v \leq \overline{\mathbf{W}}_v \mathbf{x} + \overline{\mathbf{b}}_v = \overline{h}(\mathbf{x}, \mathbf{u}_v), \qquad (5)$$

where $\underline{\mathbf{W}}_v = \overline{\mathbf{W}}_v = \nabla_{\mathbf{x}}^\top h(\mathbf{x}, \mathbf{u}_v)$ and $\underline{\mathbf{b}}_v, \overline{\mathbf{b}}_v$ are shown in Remark 1 in Appendix A.2 through Lagrange remainder with bounded $\ell_2$ operator norm of Hessian matrix for each entry of $h(\mathbf{x}, \mathbf{u}_v)$, following [45].

### 3.2 Bounding the Jacobian of Neural CBF

Now we consider the gradient of neural CBF $\phi : \mathbb{R}^n \to \mathbb{R}$. As shown in Section 2.1, neural CBFs alternatively consist of linear layers $\hat{\mathbf{z}}_i = y_i(\mathbf{z}_{i-1}) = \mathbf{W}_i \mathbf{z}_{i-1} + \mathbf{b}_i$ and ReLU activation layers $\mathbf{z}_{i-1} = \sigma_{i-1}(\hat{\mathbf{z}}_{i-1})$, with input and output as $\mathbf{z}_0 = \mathbf{x}, \hat{\mathbf{z}}_L = \phi(\mathbf{x})$. For the gradients of $\phi$ with respect to intermediate vectors $\mathbf{z}_{i-1}, \hat{\mathbf{z}}_i, \forall i = 1, 2, \ldots, L$, based on chain rule, we have

$$\nabla_{\mathbf{z}_{i-1}}\phi = \nabla_{\mathbf{z}_{i-1}}^\top y_i \nabla_{\hat{\mathbf{z}}_i}\phi = \mathbf{W}_i^\top \nabla_{\hat{\mathbf{z}}_i}\phi, \ \nabla_{\hat{\mathbf{z}}_i}\phi = \frac{\partial \mathbf{z}_i}{\partial \hat{\mathbf{z}}_i} \odot \nabla_{\mathbf{z}_i}\phi = \sigma'(\hat{\mathbf{z}}_i) \odot \nabla_{\mathbf{z}_i}\phi, \qquad (6)$$

where $\odot$ denotes element-wise Hadamard product for two inputs with the same size. Based on the initial condition $\nabla_{\hat{\mathbf{z}}_L}\phi = \mathbf{1}$, the recursive formula (6) will result in the gradient w.r.t the input $\nabla_{\mathbf{x}}\phi = \nabla_{\mathbf{z}_0}\phi$ in a back-propagation manner. More specifically, with initial trivial linear bounds of $\mathbf{0}\mathbf{x} + \mathbf{W}_L^\top \leq \nabla_{\mathbf{z}_{L-1}}\phi \leq \mathbf{0}\mathbf{x} + \mathbf{W}_L^\top$, the final gradient $\nabla_{\mathbf{x}}\phi$ can be linearly bounded as

$$\underline{\nabla}(\mathbf{x}) = \underline{\mathbf{\Lambda}}\mathbf{x} + \underline{\mathbf{d}} \leq \nabla_{\mathbf{x}}\phi \leq \overline{\nabla}(\mathbf{x}) = \overline{\mathbf{\Lambda}}\mathbf{x} + \overline{\mathbf{d}}, \text{ where } \underline{\mathbf{\Lambda}} = \overline{\mathbf{\Lambda}} \equiv \mathbf{0} \text{ and for } j\text{-th entry of } \nabla_{\mathbf{x}}\phi, \quad (7)$$

$$[\underline{\mathbf{d}}]_j = \min_{\mathbf{x} \in \Delta\mathcal{X}} \left[ [\prod_{i=1}^{L-1} \mathbf{W}_i^\top \text{Diag}(\sigma'(\hat{\mathbf{z}}_i))] \mathbf{W}_L^\top \right]_j, \ [\overline{\mathbf{d}}]_j = \max_{\mathbf{x} \in \Delta\mathcal{X}} \left[ [\prod_{i=1}^{L-1} \mathbf{W}_i^\top \text{Diag}(\sigma'(\hat{\mathbf{z}}_i))] \mathbf{W}_L^\top \right]_j.$$

We remark that the bound above is essentially equivalent to the linear bounds in [46] because the initial gradient of the last dense layer $\nabla_{\hat{\mathbf{z}}_L}\phi = \mathbf{1}$ is not related to input $\mathbf{x}$ and each derivative of ReLU activation function $\sigma'(\hat{\mathbf{z}}_i)$ is $0 - 1$ bounded Heaviside step function. Therefore, the input $\mathbf{x}$ in the pre-activation bound $\hat{\mathbf{z}}_i$ will never appear in the bound propagation of $\nabla_{\mathbf{x}}\phi$, resulting in the weight term with respect to $\mathbf{x}$ always being $\mathbf{0}$. Equation (7) greatly simplifies linear bound propagation through the inner product, as shown in the next section. We give the following simple example to show the Jacobian and optimal control input.

**Example 1.** *Consider 1D double-integrator* $\dot{\mathbf{x}} = \begin{bmatrix} 0 & 1 \\ 0 & 0 \end{bmatrix}\mathbf{x} + \begin{bmatrix} 0 \\ 1 \end{bmatrix}\mathbf{u}$ *with* $\begin{bmatrix} -0.1 \\ -0.1 \end{bmatrix} \leq \mathbf{x} \leq \begin{bmatrix} 0 \\ 0.1 \end{bmatrix}$ *and* $-1 \leq \mathbf{u} \leq 1$, *given a 2-layer neural CBF* $\phi(\mathbf{x}) = \begin{bmatrix} 1 & 1 \end{bmatrix} ReLU(\begin{bmatrix} \sqrt{2} & 1 \\ \sqrt{2} & -1 \end{bmatrix}\mathbf{x}) - 0.05$, *based on Equation (7), the linear bounds of* $\nabla_{\mathbf{x}}\phi(\mathbf{x})$ *are* $\mathbf{0}\mathbf{x} + \begin{bmatrix} 0 \\ -1 \end{bmatrix} \leq \nabla_{\mathbf{x}}\phi \leq \mathbf{0}\mathbf{x} + \begin{bmatrix} \sqrt{2} \\ 1 \end{bmatrix}$. *Also, the optimal control input from Equation (4) is* $\mathbf{u}_v = -1$ *when* $\begin{bmatrix} \sqrt{2} & 1 \end{bmatrix}\mathbf{x} > 0$ *and* $\mathbf{u}_v = 1$ *when* $\begin{bmatrix} \sqrt{2} & -1 \end{bmatrix}\mathbf{x} > 0$.

### 3.3 Verification with Symbolic Bound Propagation through Inner Product

By solving the inner minimization w.r.t control input $\mathbf{u}$ based on Proposition 1, the equivalent verification goal of $\max_{\mathbf{x} \in \Delta\mathcal{X}_\phi} \min_{\mathbf{u} \in \mathcal{U}} \dot{\phi}(\mathbf{x}, \mathbf{u}) + \alpha\phi(\mathbf{x})$ in Equation (3) is

$$\max_{\mathbf{x} \in \Delta\mathcal{X}_\phi} \dot{\phi}(\mathbf{x}, \mathbf{u}_v) + \alpha\phi(\mathbf{x}) = \max_{\mathbf{x} \in \Delta\mathcal{X}_\phi} \nabla_{\mathbf{x}}\phi^\top h(\mathbf{x}, \mathbf{u}_v) + \alpha\phi(\mathbf{x}) \leq 0, \qquad (8)$$

which contains the inner product of bounded neural CBF Jacobian $\nabla_{\mathbf{x}}\phi$ in Equation (7) and linearly bounded system dynamics $h(\mathbf{x}, \mathbf{u}_v)$ in Equation (5). We define the problem of bound propagation through the inner product below.

**Problem 1.** *Given two vector functions $f_1(\mathbf{x}), f_2(\mathbf{x}) : \mathbb{R}^n \to \mathbb{R}^m$ with linear symbolic bounds w.r.t input $\mathbf{x} \in \mathbb{R}^n$, i.e. $\underline{\mathbf{W}}_1\mathbf{x}+\underline{\mathbf{b}}_1 \leq f_1(\mathbf{x}) \leq \overline{\mathbf{W}}_1\mathbf{x}+\overline{\mathbf{b}}_1, \underline{\mathbf{W}}_2\mathbf{x}+\underline{\mathbf{b}}_2 \leq f_2(\mathbf{x}) \leq \overline{\mathbf{W}}_2\mathbf{x}+\overline{\mathbf{b}}_2$, the problem is to find the linear symbolic bounds for the inner product of $f_1(x), f_2(\mathbf{x})$, i.e. find $\underline{\mathbf{W}}_p, \underline{\mathbf{b}}_p, \overline{\mathbf{W}}_p, \overline{\mathbf{b}}_p$ s.t. $\underline{\mathbf{W}}_p\mathbf{x} + \underline{\mathbf{b}}_p \leq \langle f_1(\mathbf{x}), f_2(\mathbf{x}) \rangle = f_1^\top(\mathbf{x})f_2(\mathbf{x}) \leq \overline{\mathbf{W}}_p\mathbf{x} + \overline{\mathbf{b}}_p$.*

In our setting of Equation (8), we have $f_1(\mathbf{x}) = \nabla_\mathbf{x}\phi, f_2(\mathbf{x}) = h(\mathbf{x}, \mathbf{u}_v)$. To the best of our knowledge, it is not trivial to solve the problem above. However, with the Jacobian of ReLU-based $\phi$ where $\underline{\mathbf{\Lambda}} = \overline{\mathbf{\Lambda}} \equiv \mathbf{0}$ in Equation (7), the problem will be simplified and more straightforward to solve. We derive the following theorem as a sound upper bound of the left-hand side of Equation (8) to verify our goal in Equation (3). Proof can be found in Appendix A.2.

**Theorem 2.** *For any control-affine system $h(\mathbf{x}, \mathbf{u}) = f(\mathbf{x}) + g(\mathbf{x})\mathbf{u}$ with bounded control input $\mathbf{u} \in \mathcal{U}$, given a learned neural CBF $\phi(\mathbf{x})$ with ReLU activation functions, suppose the boundary state set $\partial\mathcal{X}_\phi$ is the union of $K$ hyper-rectangles $\Delta\mathcal{X}$ as $\partial\mathcal{X}_\phi \subset \Delta\mathcal{X}_\phi := \cup_{k=1}^K \Delta\mathcal{X}^{(k)}$, then the goal in Equation (3) is achieved if as a sound upper bound of $\nabla_\mathbf{x}\phi^\top h(\mathbf{x}, \mathbf{u}_v) + \alpha\phi(\mathbf{x})$, the following inequality holds for any $\mathbf{x}$ in each hyper-rectangle state set $\Delta\mathcal{X} \in \Delta\mathcal{X}_\phi$,*

$$[\overline{\mathbf{d}}^\top]_+[\overline{h}(\mathbf{x}, \mathbf{u}_v)]_+ + [\underline{\mathbf{d}}^\top]_+[\overline{h}(\mathbf{x}, \mathbf{u}_v)]_- + [\overline{\mathbf{d}}^\top]_-[\underline{h}(\mathbf{x}, \mathbf{u}_v)]_+ + [\underline{\mathbf{d}}^\top]_-[\underline{h}(\mathbf{x}, \mathbf{u}_v)]_- + \alpha\phi(\mathbf{x}) \leq 0,$$

*where $[*]_+ = \max\{0, *\} = ReLU(*)$, $[*]_- = \min\{0, *\} = -ReLU(-*)$, and $\overline{\mathbf{d}}, \underline{\mathbf{d}}$ and $\overline{h}(\mathbf{x}, \mathbf{u}_v), \underline{h}(\mathbf{x}, \mathbf{u}_v)$ can be found through Equation (7) and Equation (5), respectively.*

If $\overline{h}(\mathbf{x}, \mathbf{u}_v), \underline{h}(\mathbf{x}, \mathbf{u}_v)$ are concretized and non-symbolic, our result degrades to the interval arithmetic case [47, 36], showing our symbolic bounds are more general and tighter. Note that for efficient implementation, given $\Delta\mathcal{X}$, the optimal control input $\mathbf{u}$ is approximated as fixed by Proposition 1. The symbolic upper bound of Theorem 2 can be viewed as the summation of new MLPs with ReLU activation functions, which can be verified through CROWN [38, 39]. Combining branch-and-bound scheme [48, 36, 17], the new optimal control input $\mathbf{u}'$ can be found for each new split branch $\Delta\mathcal{X}'$ for the recursive verification. Based on Example 1, we continue to find the corresponding symbolic upper bound of Theorem 2 with branch-and-bound below.

**Example 2.** *To verify Example 1 with $\alpha = 0.5$, initially with $\begin{bmatrix} -0.1 \\ -0.1 \end{bmatrix} \leq \mathbf{x} \leq \begin{bmatrix} 0 \\ 0.1 \end{bmatrix}$, the verification condition in Theorem 2 with approximated optimal control input $\mathbf{u}_v = -1$ is $\begin{bmatrix} \sqrt{2} & 1 \end{bmatrix} ReLU(\begin{bmatrix} 0 & 1 \\ 0 & 0 \end{bmatrix} \mathbf{x} + \begin{bmatrix} 0 \\ -1 \end{bmatrix}) - \begin{bmatrix} 0 & -1 \end{bmatrix} ReLU(-\begin{bmatrix} 0 & 1 \\ 0 & 0 \end{bmatrix} \mathbf{x} - \begin{bmatrix} 0 \\ -1 \end{bmatrix}) + 0.5 \times ([\begin{bmatrix} 1 & 1 \end{bmatrix} ReLU(\begin{bmatrix} \sqrt{2} & 1 \\ \sqrt{2} & -1 \end{bmatrix} \mathbf{x}) - 0.05) \leq 0$, which cannot be verified directly. Then we further split $\mathbf{x}$ specification into two new branches $\begin{bmatrix} -0.1 \\ -0.1 \end{bmatrix} \leq \mathbf{x} \leq \begin{bmatrix} 0 \\ 0 \end{bmatrix}$ and $\begin{bmatrix} -0.1 \\ 0 \end{bmatrix} \leq \mathbf{x} \leq \begin{bmatrix} 0 \\ 0.1 \end{bmatrix}$, and the verification condition of the first branch is $\begin{bmatrix} \sqrt{2} & 0 \end{bmatrix} ReLU(\begin{bmatrix} 0 & 1 \\ 0 & 0 \end{bmatrix} \mathbf{x} + \begin{bmatrix} 0 \\ 1 \end{bmatrix}) - \begin{bmatrix} 0 & -1 \end{bmatrix} ReLU(-\begin{bmatrix} 0 & 1 \\ 0 & 0 \end{bmatrix} \mathbf{x} - \begin{bmatrix} 0 \\ 1 \end{bmatrix}) + 0.5 \times ([\begin{bmatrix} 0 & -1 \end{bmatrix} ReLU(\begin{bmatrix} \sqrt{2} & 1 \\ \sqrt{2} & -1 \end{bmatrix} \mathbf{x}) - 0.05) \leq 0$ with $\mathbf{u}_v = 1$, and the verification condition of the second branch is $\begin{bmatrix} \sqrt{2} & 1 \end{bmatrix} ReLU(\begin{bmatrix} 0 & 1 \\ 0 & 0 \end{bmatrix} \mathbf{x} + \begin{bmatrix} 0 \\ -1 \end{bmatrix}) - \begin{bmatrix} 0 & 0 \end{bmatrix} ReLU(-\begin{bmatrix} 0 & 1 \\ 0 & 0 \end{bmatrix} \mathbf{x} - \begin{bmatrix} 0 \\ -1 \end{bmatrix}) + 0.5 \times ([\begin{bmatrix} 0 & -1 \end{bmatrix} ReLU(\begin{bmatrix} \sqrt{2} & 1 \\ \sqrt{2} & -1 \end{bmatrix} \mathbf{x}) - 0.05) \leq 0$ with $\mathbf{u}_v = -1$, which can be verified or further split through off-the-shelf verification tools [46] following such branch-and-bound scheme.*

## 4 Experiments

In this section, we aim to answer the following questions. Given pre-trained neural CBFs, how does our proposed verification method perform compared to the existing verification baseline under different robot dynamics? In terms of tightness and scalability, how are our proposed method and the baselines influenced by different input specification sizes and model complexity? We answer the

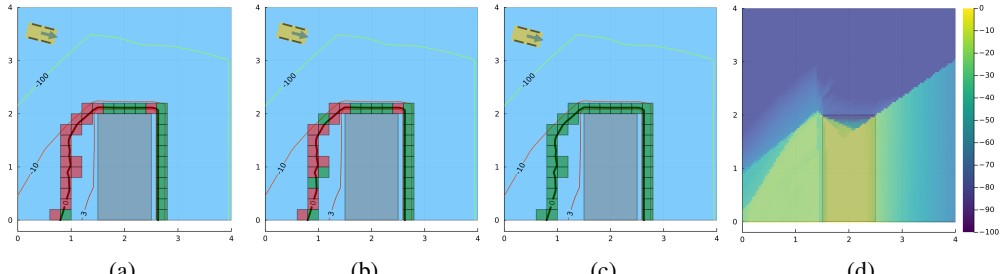

|  | (a) | (b) | (c) | (d) |

Figure 2: Illustration of the verification results for Dubins car using (a) NNCB-IBP [28], (b) BBV [36] and (c) ours. The obstacle is in gray. The state boundary of $\phi(\mathbf{x}) = 0$ given orientation $-14°$ is over-approximated by boxes. Green boxes denote the results of "hold" while red ones denote the results of "unknown" to verify $\dot{\phi} + 0.5\phi \leq 0$, whose value is shown as heatmap in Fig. (d).

| CBF training type | Regular training | | | Adversarial training | | |
| Verification condition | $\alpha = 0.1$ | $\alpha = 0.5$ | $\alpha = 1.0$ | $\alpha = 0.1$ | $\alpha = 0.5$ | $\alpha = 1.0$ |
|---|---|---|---|---|---|---|
| **Dubins Car** 3-dim state | | | | | | |
| BBV [36] | 0.617 | 0.492 | 0.448 | 0.750 | 0.719 | 0.695 |
| NNCB-IBP [28] | 0.446 | 0.437 | 0.425 | 0.623 | 0.601 | 0.577 |
| Ours | **0.700** | **0.729** | **0.706** | **0.778** | **0.730** | **0.732** |
| **Point Robot** 4-dim state | | | | | | |
| BBV [36] | 0.384 | 0.355 | 0.317 | 0.391 | 0.367 | 0.337 |
| NNCB-IBP [28] | 0.280 | 0.261 | 0.237 | 0.317 | 0.304 | 0.282 |
| Ours | **0.404** | **0.401** | **0.391** | **0.404** | **0.397** | **0.385** |
| **Planar Quadrotor** 6-dim state | | | | | | |
| BBV [36] | 0.314 | 0.319 | 0.290 | 0.352 | 0.335 | 0.313 |
| NNCB-IBP [28] | 0.301 | 0.279 | 0.249 | 0.216 | 0.205 | 0.168 |
| Ours | **0.354** | **0.382** | **0.341** | **0.359** | **0.404** | **0.475** |

Table 1: Comparison of Verified Rate with baselines on different dynamics models under different neural CBF training settings.

first question in Section 4.2 through quantitative and qualitative comparison and answer the second one in Section 4.3 as the ablation study. Prior to that, we first introduce the experiment setup of robot dynamics and verification details.

## 4.1 Experimental Setup

**Robot dynamics and neural CBF training** We conduct experiments using three different robot dynamics, Point Robot [49, 36] with state dimension of 4, Dubins Car [50] with state dimension of 3 and Planar Quadrotor [51] with state dimension of 6. Following [36], for any random trajectories with bounded control input and state, the robots should avoid obstacles in a non-convex environment as shown in Figure 2. To this end, the neural CBF is learned to ensure forward invariance of the collision-free states. Similar to [17], for each robot dynamics, we train the neural CBF based on [14] as *regular training* and further adopt gradient-based *adversarial training* [37, 52] to enhance the forward invariance under worst-case state [18]. As *default* models, we adopt 4-layer MLPs with ReLU with layer dimensions of (16,64,16,1) to model neural CBFs, and further investigate *small* model with layer dimensions of (8,8,8,1) and *large* models with layer dimensions of (64,128,64,1) in ablation study of model complexity. More details can be found in Appendix B.1.

**Verification details and evaluation metric.** Given the pre-trained neural CBF, the goal is to verify the forward invariance in Equation (3) over the boundary states, which can be found through grid search as all the hyper-rectangles that contain zero roots of $\phi(\mathbf{x}) = 0$ following [35, 18]. The default number of grids for each dimension is 20. By traversing all boundary hyper-rectangles, we report the ratio of hyper-rectangles that satisfy the verification condition of Equation (3) as the evaluation metric *Verified Rate*. We choose two recent incomplete verification methods as the baselines, NNCB-IBP in the non-stochastic continuous version adapted from [28] and BBV [36], which are both based on interval arithmetic through CROWN bounds [38, 39] and the latter adopts branch-and-bound scheme. For a fair comparison, ours adopts the same maximum splitting iteration number in the branch-and-bound scheme as BBV. Also, following [53, 54, 14], we adopt efficient sampling-based

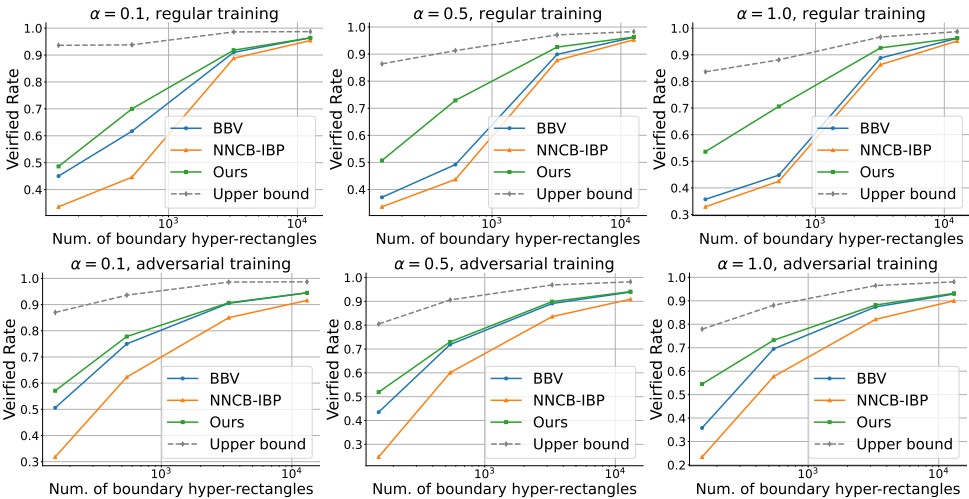

Figure 3: Verified Rate under different number of boundary hyper-rectangles under different grid sizes. The grid number per dimension for each figure (from left to right): 10, 20, 50, 100. The dashed line denotes the results of falsification by counterexample as upper bounds.

counterexample falsification to obtain an (unsound) upper bound for the verified rate [37]. We remark that since we purely focus on verification given pre-trained CBFs in this work, we do not fully explore CBF training potential and the performance of Verified Rate can be further improved by combining verification-in-the-loop training [36, 55], counterexample falsification [27, 40], etc.

## 4.2  Results Comparison

**Quantitative results.**  From Table 1, we can see that under different robot dynamics, ours performs better than the baselines, although the verification becomes more challenging to scale up for higher state dimensions. Specifically, ours and BBV [36] have better performance compared to NNCB-IBP [28] due to the branch-and-bound scheme. With larger $\alpha$ in verification condition (3), the performance of the baseline method decays significantly due to larger over-approximation errors by interval arithmetic, while ours is tighter and keeps relatively higher Verified Rate. The advantage of our method over baselines is larger for neural CBFs with regular training than for neural CBFs with adversarial training, showing adversarial training can enforce the forward invariance to hold.

**Qualitative results.**  In this section, we visualize the verification results of Dubins Car for qualitative comparison with baselines in Figure 2. Given a fixed orientation angle, the boundary of $\phi(\mathbf{x}) = 0$ is over-approximated as position boxes as input specifications. It can be seen that it is more challenging for both baselines [28, 36] in Figure 2(a,b) to verify the boxes that face the car head, which are intuitively more likely to result in collision. However, due to the tighter bounds of our symbolic propagation during verification, ours can verify all the boundary boxes, which is consistent with the heatmap visualization of the boundary condition values in Equation (2).

## 4.3  Ablation Study

**Different numbers of boundary hyper-rectangles under different grid sizes.**  Since the boundary hyper-rectangles are found through grid search with fixed grid size, the finer the grids are, the more boundary hyper-rectangles will be. As shown in Figure 3, we evaluate the performance with different fine-grained boundaries from 10, 20, 50, and 100 grids for each dimension, respectively. It can be seen that the performance of both ours and baselines increases as the number of boundary hyper-rectangles goes up, but ours consistently presents higher Verified Rate than the baselines do. More specifically, with fewer boundary hyper-rectangles, our superiority over the baselines becomes more significant even for branch-and-bound based BBV [36], validating the efficiency of our method. Although there is still a margin with respect to the unsound upper bound, our performance can be further boosted through complete verification methods [48] for tighter bounds.

| Verified rate of Dubins Car | | Regular training | | | Adversarial training | | |
|---|---|---|---|---|---|---|---|
| under different model sizes | | Small | Default | Large | Small | Default | Large |
| $\alpha = 0.1$ | BBV [36] | 0.506 | 0.617 | 0.729 | 0.785 | 0.750 | 0.458 |
| | Ours | **0.558** | **0.700** | **0.752** | **0.788** | **0.778** | **0.481** |
| $\alpha = 0.5$ | BBV [36] | 0.365 | 0.492 | 0.635 | 0.781 | 0.719 | 0.411 |
| | Ours | **0.562** | **0.729** | **0.688** | **0.792** | **0.730** | **0.444** |
| $\alpha = 1.0$ | BBV [36] | 0.208 | 0.448 | 0.510 | 0.750 | 0.695 | 0.367 |
| | Ours | **0.540** | **0.706** | **0.592** | **0.758** | **0.732** | **0.413** |

Table 2: The influence of neural CBFs with different model sizes on branch-and-bound based verification baseline and ours with Dubins Car. The Small, Default and Large denote 4-layer ReLU-MLPs with layer output dimensions of (8,8,8,1), (16,64,16,1) and (64,128,64,1), respectively.

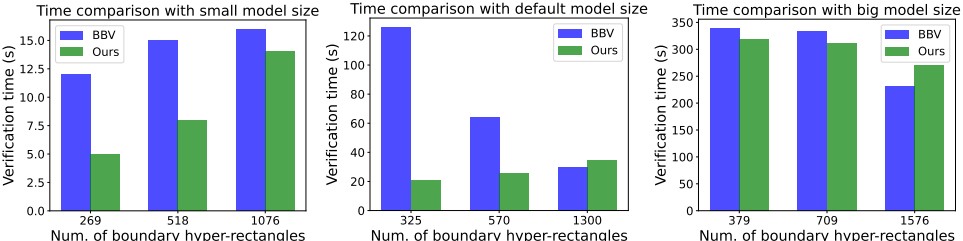

Figure 4: Comparison of verification time using branch-and-bound to achieve 100% Verified Rate with different model complexity with regular training. The more hyper-rectangles the boundary set contains, the finer the state input specification will be.

**Different complexity of neural CBF models.** As shown in Table 2, we investigate the verification performance under different sizes of neural CBFs under regular training and adversarial training. We can see that for regular training, as models become larger, the verified rate will mostly increase due to better overall performance, especially for the baseline BBV [36]. However, since even small models can be effectively robustified and obtain good overall performance through adversarial training, the verified rate tends to decrease as model complexity increases because larger models are more difficult to verify. Besides, , ours can consistently give higher verified rates compared to the baseline.

**Verification time of branch-and-bound.** For a fair comparison of time consumption, we allow branch-and-bound to run forever until achieving 100% verified rate of $\alpha = 0.1$ under Dubins Car with regular training . From Figure 4, we can see that with small model size, the total time along the boundary is mainly dominated by the number of boundary hyper-rectangle inputs, and ours performs faster on small NNs. However, with a large model size, the total time is mainly dominated by the single-input verification time and the more hyper-rectangles along the boundary, the finer the input spec will be and the faster the single-input verification will be . In this case, ours cannot always beat BBV since the non-symbolic interval arithmetic is more computationally efficient for single-input verification with a larger model size. In between, the default model size tells us that the total verification time for ours has the bottleneck of the number of hyper-rectangles, while the baseline has the bottleneck of single-input verification time.

## 5 Conclusion

In this work, we propose an efficient verification framework for neural CBFs using symbolic bound propagation, which combines the linearly bounded nonlinear dynamic system and the linear bounds of neural CBFs Jacobian. By utilizing bounded derivatives of activation functions in neural networks, we demonstrate that linear symbolic bounds can be propagated through the inner product of the neural CBF gradient and nonlinear system dynamics, fully keeping the symbolic bound for better tightness. Extensive experiments on various robot dynamics show that our method outperforms interval arithmetic based baselines in terms of verified rate and verification time along the CBF boundary with different model complexity. These results validate the tightness and efficiency of our proposed method.

**Acknowledgments**

This work is in part supported by the National Science Foundation under Grant No. 2144489.

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

# A Proofs

## A.1 Proof of Proposition 1

**Lemma 1** (Interval Arithmetic, restated from Section 4.1 in [47].). *For any matrix multiplication* $\mathbf{A} \cdot \mathbf{x} : \mathbb{R}^n \to \mathbb{R}^m$, *if* $\mathbf{x}$ *is entry-wisely bounded as* $\underline{\mathbf{x}} \leq \mathbf{x} \leq \overline{\mathbf{x}}$, *i.e.* $\underline{\mathbf{x}}_i \leq \mathbf{x}_i \leq \overline{\mathbf{x}}_i, \forall i = 1 \ldots n$, *the following inequalities hold for each entry of* $\mathbf{A}\mathbf{x}$,

$$[\mathbf{A}]_+ \underline{\mathbf{x}} + [\mathbf{A}]_- \overline{\mathbf{x}} \leq \mathbf{A}\mathbf{x} \leq [\mathbf{A}]_- \underline{\mathbf{x}} + [\mathbf{A}]_+ \overline{\mathbf{x}} \tag{9}$$

*where* $[\cdot]_+ := \max\{0, \cdot\}, [\cdot]_- := \min\{0, \cdot\}$.

*Proof.* For the lower bound $[\mathbf{A}]_+ \underline{\mathbf{x}} + [\mathbf{A}]_- \overline{\mathbf{x}}$, consider the $j$-th entry of $[\mathbf{A}\mathbf{x}]_j = \sum_{i=1}^n \mathbf{A}_{j,i} \mathbf{x}_i$. With the entry-wise bounds $\underline{\mathbf{x}}_i \leq \mathbf{x}_i \leq \overline{\mathbf{x}}_i, \forall i = 1 \ldots n$, if $\mathbf{A}_{j,i} \geq 0$, it holds that $\mathbf{A}_{j,i} \underline{\mathbf{x}}_i \leq \mathbf{A}_{j,i} \mathbf{x}_i$; similarly, if $\mathbf{A}_{j,i} < 0$, it holds that $\mathbf{A}_{j,i} \overline{\mathbf{x}}_i \leq \mathbf{A}_{j,i} \mathbf{x}_i$. Writing it compactly, we have

$$[\mathbf{A}_{j,i}]_+ \underline{\mathbf{x}}_i + [\mathbf{A}_{j,i}]_- \overline{\mathbf{x}}_i = \max\{0, \mathbf{A}_{j,i}\} \underline{\mathbf{x}}_i + \min\{0, \mathbf{A}_{j,i}\} \overline{\mathbf{x}}_i \leq \mathbf{A}_{j,i} \mathbf{x}_i$$

. By summing the inequality above over $i = 1, \ldots, n$, it holds that

$$[[\mathbf{A}]_+ \underline{\mathbf{x}}]_j + [[\mathbf{A}]_- \overline{\mathbf{x}}]_j = \sum_{i=1}^n [\mathbf{A}_{j,i}]_+ \underline{\mathbf{x}}_i + \sum_{i=1}^n [\mathbf{A}_{j,i}]_- \overline{\mathbf{x}}_i \leq [\mathbf{A}\mathbf{x}]_j = \sum_{i=1}^n \mathbf{A}_{j,i} \mathbf{x}_i$$

, which indicates $[\mathbf{A}]_+ \underline{\mathbf{x}} + [\mathbf{A}]_- \overline{\mathbf{x}} \leq \mathbf{A}\mathbf{x}$ holds for each entry $j$. Similarly, the upper bound $\mathbf{A}\mathbf{x} \leq [\mathbf{A}]_- \underline{\mathbf{x}} + [\mathbf{A}]_+ \overline{\mathbf{x}}$ can be derived in the same way, concluding the proof the interval arithmetic. $\square$

**Proposition 2.** *(restated of Proposition 1) When* $\mathbf{u}$ *is within a hyper-rectangle* $\mathcal{U} = [\underline{\mathbf{u}}, \overline{\mathbf{u}}] = \{\mathbf{u} \in \mathbb{R}^m \mid \underline{\mathbf{u}} \leq \mathbf{u} \leq \overline{\mathbf{u}}\}$, *given* $\mathbf{x} \in \mathcal{X}$, *the minimum value of* $\dot{\phi}(\mathbf{x}, \mathbf{u})$ *over* $\mathbf{u} \in \mathcal{U}$ *can be found explicitly as,*

$$\min_{\mathbf{u} \in \mathcal{U}} \dot{\phi}(\mathbf{x}, \mathbf{u}) = \nabla_{\mathbf{x}} \phi^\top f(\mathbf{x}) + [\nabla_{\mathbf{x}} \phi^\top g(\mathbf{x})]_+ \underline{\mathbf{u}} + [\nabla_{\mathbf{x}} \phi^\top g(\mathbf{x})]_- \overline{\mathbf{u}} = \dot{\phi}(\mathbf{x}, \mathbf{u}_v(\mathbf{x})), \tag{10}$$

*where* $[*]_+ = \max\{0, *\}, [*]_- = \min\{0, *\}$ *and the optimal control input* $\mathbf{u}_v(\mathbf{x}) = \arg\min_{\mathbf{u} \in V(\mathcal{U})} \dot{\phi}(\mathbf{x}, \mathbf{u})$ *lies among the vertices* $V(\mathcal{U})$ *of hyper-rectangle* $\mathcal{U}$ *given state* $\mathbf{x}$.

*Proof.* Based on the chain rule, it holds that

$$\dot{\phi}(\mathbf{x}, \mathbf{u}) = \nabla_{\mathbf{x}} \phi^\top \dot{\mathbf{x}} = \nabla_{\mathbf{x}} \phi^\top f(\mathbf{x}) + \nabla_{\mathbf{x}} \phi^\top g(\mathbf{x}) \mathbf{u}.$$

With $\underline{\mathbf{u}} \leq \mathbf{u} \leq \overline{\mathbf{u}}$, based on Lemma 1, it holds that

$$[\nabla_{\mathbf{x}} \phi^\top g(\mathbf{x})]_+ \underline{\mathbf{u}} + [\nabla_{\mathbf{x}} \phi^\top g(\mathbf{x})]_- \overline{\mathbf{u}} \leq \nabla_{\mathbf{x}} \phi^\top g(\mathbf{x}) \mathbf{u}.$$

Besides, consider the vertices $V(\mathcal{U}) := \{\mathbf{u}_v \mid [\mathbf{u}_v]_i \in \{\underline{\mathbf{u}}_i, \overline{\mathbf{u}}_i\}, \forall i = 1, \ldots, m\}$, we can find the equivalent expression for the lower bounds,

$$[\nabla_{\mathbf{x}} \phi^\top g(\mathbf{x})]_+ \underline{\mathbf{u}} + [\nabla_{\mathbf{x}} \phi^\top g(\mathbf{x})]_- \overline{\mathbf{u}} = \nabla_{\mathbf{x}} \phi^\top g(\mathbf{x}) \mathbf{u}_v(\mathbf{x}),$$

where $[\mathbf{u}_v(\mathbf{x})]_i = \underline{\mathbf{u}}_i$ if $[\nabla_{\mathbf{x}} \phi^\top g(\mathbf{x})]_i \geq 0$ and $[\mathbf{u}_v(\mathbf{x})]_i = \overline{\mathbf{u}}_i$ if $[\nabla_{\mathbf{x}} \phi^\top g(\mathbf{x})]_i < 0$, showing that there exists $\mathbf{u}_v \in V(\mathcal{U})$ s.t. the lower bound $[\nabla_{\mathbf{x}} \phi^\top g(\mathbf{x})]_+ \underline{\mathbf{u}} + [\nabla_{\mathbf{x}} \phi^\top g(\mathbf{x})]_- \overline{\mathbf{u}}$ can be achieved equivalently, which concludes that proof. $\square$

## A.2 Proof of Theorem 2

**Theorem 3.** *(restated of Theorem 2.) For any control-affine system* $h(\mathbf{x}, \mathbf{u}) = f(\mathbf{x}) + g(\mathbf{x})\mathbf{u}$ *with bounded control input* $\mathbf{u} \in \mathcal{U}$, *given a learned neural CBF* $\phi(\mathbf{x})$ *with ReLU activation functions, suppose the boundary state set* $\partial \mathcal{X}_\phi$ *is the union of* $K$ *hyper-rectangles* $\Delta \mathcal{X}$ *as* $\partial \mathcal{X}_\phi \subset \Delta \mathcal{X}_\phi :=$ $\cup_{k=1}^K \Delta \mathcal{X}^{(k)}$, *then the following inequality* $\max_{\mathbf{x} \in \partial \mathcal{X}_\phi} \min_{\mathbf{u} \in \mathcal{U}} \dot{\phi}(\mathbf{x}, \mathbf{u}) + \alpha \phi(\mathbf{x}) \leq 0$ *is satisfied if as a sound upper bound of* $\nabla_{\mathbf{x}} \phi^\top h(\mathbf{x}, \mathbf{u}_v) + \alpha \phi(\mathbf{x})$, *the following inequality holds for any* $\mathbf{x}$ *in each hyper-rectangle state set* $\Delta \mathcal{X} \in \Delta \mathcal{X}_\phi$,

$$[\overline{\mathbf{d}}^\top]_+ [\overline{h}(\mathbf{x}, \mathbf{u}_v)]_+ + [\underline{\mathbf{d}}^\top]_+ [\overline{h}(\mathbf{x}, \mathbf{u}_v)]_- + [\overline{\mathbf{d}}^\top]_- [\underline{h}(\mathbf{x}, \mathbf{u}_v)]_+ + [\underline{\mathbf{d}}^\top]_- [\underline{h}(\mathbf{x}, \mathbf{u}_v)]_- + \alpha \phi(\mathbf{x}) \leq 0,$$

*where* $[*]_+ = \max\{0, *\} = ReLU(*)$, $[*]_- = \min\{0, *\} = -ReLU(-*)$, *and* $\overline{\mathbf{d}}, \underline{\mathbf{d}}$ *and* $\overline{h}(\mathbf{x}, \mathbf{u}_v), \underline{h}(\mathbf{x}, \mathbf{u}_v)$ *are the lower and upper bounds of* $\nabla_{\mathbf{x}} \phi$ *and* $h(\mathbf{x}, \mathbf{u}_v)$, *respectively.*

*Proof.* Based on Proposition 2, we have the following inequality hold for any $\Delta\mathcal{X} \in \Delta\mathcal{X}_\phi$,

$$\max_{\mathbf{x}\in\Delta\mathcal{X}} \min_{\mathbf{u}\in\mathcal{U}} \dot{\phi}(\mathbf{x},\mathbf{u}) + \alpha\phi(\mathbf{x}) = \max_{\mathbf{x}\in\Delta\mathcal{X}} \dot{\phi}(\mathbf{x},\mathbf{u}_v(\mathbf{x})) + \alpha\phi(\mathbf{x}) \le \max_{\mathbf{x}\in\Delta\mathcal{X}} \dot{\phi}(\mathbf{x},\mathbf{u}_v) + \alpha\phi(\mathbf{x}), \quad (11)$$

where $\mathbf{u}_v$ is an approximated constant vertex of optimal control input $\mathbf{u}_v(\mathbf{x})$ for a sound upper bound of $\dot{\phi}(\mathbf{x},\mathbf{u}_v(\mathbf{x}))$ over $\mathbf{x}\in\Delta\mathcal{X}$. Now with the bounded dynamics $\underline{h}(\mathbf{x},\mathbf{u}_v) \le h(\mathbf{x},\mathbf{u}_v) \le \overline{h}(\mathbf{x},\mathbf{u}_v)$, by Lemma 1, for any $\mathbf{x}\in\Delta\mathcal{X}$ we have

$$\dot{\phi}(\mathbf{x},\mathbf{u}_v) + \alpha\phi(\mathbf{x}) = \nabla_\mathbf{x}^\top \phi h(\mathbf{x},\mathbf{u}_v) + \alpha\phi(\mathbf{x}) \le [\nabla_\mathbf{x}^\top\phi]_- \underline{h}(\mathbf{x},\mathbf{u}_v) + [\nabla_\mathbf{x}^\top\phi]_+ \overline{h}(\mathbf{x},\mathbf{u}_v) + \alpha\phi(\mathbf{x}).$$

Besides, with the bounded gradient $\underline{\mathbf{d}} \le \nabla_\mathbf{x}\phi \le \overline{\mathbf{d}}$, the following inequalities hold

$$[\underline{\mathbf{d}}]_+ \le [\nabla_\mathbf{x}\phi]_+ \le [\overline{\mathbf{d}}]_+, [\underline{\mathbf{d}}]_- \le [\nabla_\mathbf{x}\phi]_- \le [\overline{\mathbf{d}}]_-.$$

Then applying Lemma 1 for $[\nabla_\mathbf{x}^\top\phi]_- \underline{h}(\mathbf{x},\mathbf{u}_v)$ and $[\nabla_\mathbf{x}^\top\phi]_+ \overline{h}(\mathbf{x},\mathbf{u}_v)$, we further have the following inequality hold for any $\mathbf{x}\in\Delta\mathcal{X}$,

$$\dot{\phi}(\mathbf{x},\mathbf{u}_v) \le [\overline{\mathbf{d}}^\top]_+ [\overline{h}(\mathbf{x},\mathbf{u}_v)]_+ + [\underline{\mathbf{d}}^\top]_+ [\overline{h}(\mathbf{x},\mathbf{u}_v)]_- + [\overline{\mathbf{d}}^\top]_- [\underline{h}(\mathbf{x},\mathbf{u}_v)]_+ + [\underline{\mathbf{d}}^\top]_- [\underline{h}(\mathbf{x},\mathbf{u}_v)]_-.$$

Therefore, if for any $\mathbf{x}$ in each hyper-rectangle state set $\Delta\mathcal{X} \in \Delta\mathcal{X}_\phi$, it holds that

$$[\overline{\mathbf{d}}^\top]_+ [\overline{h}(\mathbf{x},\mathbf{u}_v)]_+ + [\underline{\mathbf{d}}^\top]_+ [\overline{h}(\mathbf{x},\mathbf{u}_v)]_- + [\overline{\mathbf{d}}^\top]_- [\underline{h}(\mathbf{x},\mathbf{u}_v)]_+ + [\underline{\mathbf{d}}^\top]_- [\underline{h}(\mathbf{x},\mathbf{u}_v)]_- + \alpha\phi(\mathbf{x}) \le 0,$$

and then we have $\dot{\phi}(\mathbf{x},\mathbf{u}_v) + \alpha\phi(\mathbf{x}) \le 0$ for any $\mathbf{x}\in\Delta\mathcal{X}$. Combining Equation (11), we have

$$\max_{\mathbf{x}\in\Delta\mathcal{X}} \min_{\mathbf{u}\in\mathcal{U}} \dot{\phi}(\mathbf{x},\mathbf{u}) + \alpha\phi(\mathbf{x}) \le \max_{\mathbf{x}\in\Delta\mathcal{X}} \dot{\phi}(\mathbf{x},\mathbf{u}_v) + \alpha\phi(\mathbf{x}) \le 0, \forall\Delta\mathcal{X} \in \Delta\mathcal{X}_\phi.$$

Since the exact boundary of $\phi(\mathbf{x}) = 0$ is the subset of all $\Delta\mathcal{X}$, *i.e.* , $\partial\mathcal{X}_\phi \subset \Delta\mathcal{X}_\phi := \cup_{k=1}^K \Delta\mathcal{X}^{(k)}$, it holds that

$$\max_{\mathbf{x}\in\partial\mathcal{X}_\phi} \min_{\mathbf{u}\in\mathcal{U}} \dot{\phi}(\mathbf{x},\mathbf{u}) + \alpha\phi(\mathbf{x}) \le \max_{\Delta\mathcal{X}\in\Delta\mathcal{X}_\phi} \max_{\mathbf{x}\in\Delta\mathcal{X}} \min_{\mathbf{u}\in\mathcal{U}} \dot{\phi}(\mathbf{x},\mathbf{u}) + \alpha\phi(\mathbf{x}) \le 0,$$

which concludes the proof. $\square$

**Remark 1** (Linearly bounded dynamics.)**.** *We remark that although the linear bounds of dynamics* $\underline{h}(\mathbf{x},\mathbf{u}_v), \overline{h}(\mathbf{x},\mathbf{u}_v)$ *can be found through 1-order Taylor models [42, 43, 44] in practice, we can give a generally valid lower and upper bounds by assuming bounded $\ell_2$ operator norm of Hessian matrix following [45]. For the control-affine system with fixed control input* $\mathbf{u}_0, \dot{\mathbf{x}} = h(\mathbf{x},\mathbf{u}_0)$ *with bounded state* $\underline{\mathbf{x}} \le \mathbf{x} \le \overline{\mathbf{x}}$, *suppose the $\ell_2$ operator norm of Hessian matrix of $i$-th entry of* $h(\mathbf{x},\mathbf{u}_0)$ *is bounded as* $\|\nabla_\mathbf{x}^2 h^{(i)}(\mathbf{x},\mathbf{u}_0)\|_2 \le M^{(i)}$, *then at* $\mathbf{x}_0 \in [\underline{\mathbf{x}},\overline{\mathbf{x}}]$ *the following linear bounds can be found as*

$$\underline{h}(\mathbf{x},\mathbf{u}_0) = \underline{\mathbf{W}}_0 \mathbf{x} + \underline{\mathbf{b}}_0 \le h(\mathbf{x},\mathbf{u}_0) \le \overline{\mathbf{W}}_0 \mathbf{x} + \overline{\mathbf{b}}_0 = \overline{h}(\mathbf{x},\mathbf{u}_0) \text{ where } \underline{\mathbf{W}}_0 = \overline{\mathbf{W}}_0 = \nabla_\mathbf{x}^\top h(\mathbf{x}_0,\mathbf{u}_0),$$

$$\underline{\mathbf{b}}_0^{(i)} = h^{(i)}(\mathbf{x}_0,\mathbf{u}_0) - \nabla_\mathbf{x}^\top h^{(i)}(\mathbf{x}_0,\mathbf{u}_0)\mathbf{x}_0 - \frac{1}{2}\|\overline{\mathbf{x}} - \underline{\mathbf{x}}\|_2^2 M^{(i)}, \text{ for $i$-th entry of } \underline{\mathbf{b}}_0,$$

$$\overline{\mathbf{b}}_0^{(i)} = h^{(i)}(\mathbf{x}_0,\mathbf{u}_0) - \nabla_\mathbf{x}^\top h^{(i)}(\mathbf{x}_0,\mathbf{u}_0)\mathbf{x}_0 + \frac{1}{2}\|\overline{\mathbf{x}} - \underline{\mathbf{x}}\|_2^2 M^{(i)}, \text{ for $i$-th entry of } \overline{\mathbf{b}}_0.$$

*Specifically, if the control-affine system is linear and time-invariant, i.e. , $f(\mathbf{x}) = \mathbf{A}\mathbf{x}$ and $g(\mathbf{x}) = \mathbf{B}$ with constant $\mathbf{A},\mathbf{B}$, where the lower and upper bounds will trivially be* $\underline{\mathbf{W}}_0 = \overline{\mathbf{W}}_0 = \mathbf{A}, \underline{\mathbf{b}}_0 = \overline{\mathbf{b}}_0 = \mathbf{B}\mathbf{u}_0$.

## B  Experiments

In this work, symbolic bounds are bounds that are linearly related to the input specification, compared to the concretized bounds where the input dependencies are thrown away. Besides, if a sound but incomplete verifier returns a result of not-hold, maybe the statement will actually hold, but the current verifier is too loose to verify that. In contrast, if sound and complete verifiers return not-hold results, there must exist counterexamples that violate the statement. This motivates us to develop tighter sound and incomplete verification with symbolic bounds.

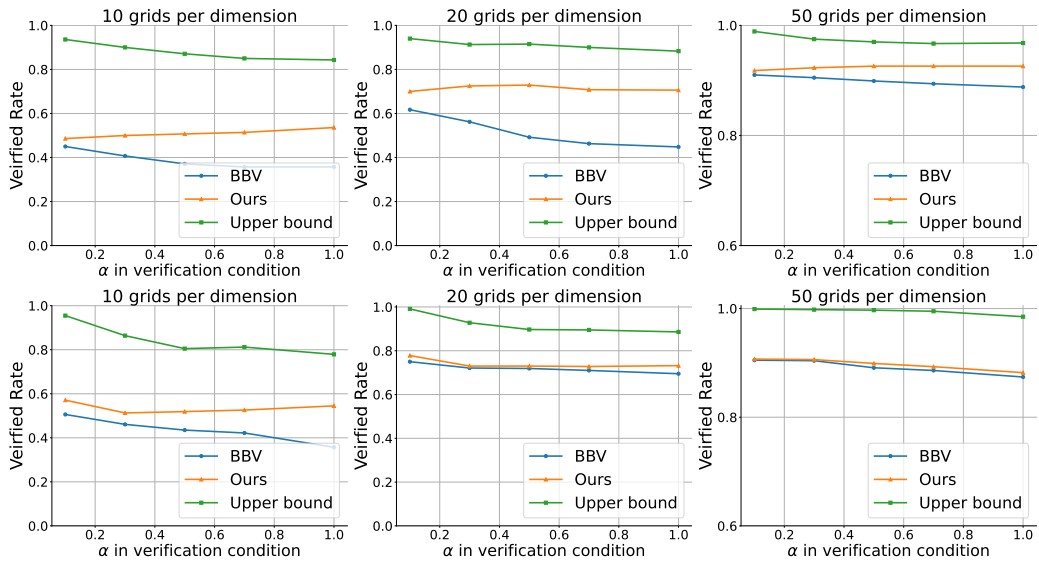

Figure 5: Verified rate with different $\alpha$ in the neural CBF verification condition using different grid sizes (different number of boundary hyper-rectangles) for Dubins Car.

## B.1 Experiment Environments and Dynamics

All the robot dynamic models are based on the open-sourced package RobotZoo.jl, where Point Robot is modified based on `DoubleIntegrator(D=2)` with zero gravity, Dubins Car is modified based on `DubinsCar` with `radius=0.175`, and Planar Quadrotor is modified based on `PlanarQuadrotor` with `mass=1.0kg`, `gravity=9.81m/s²` and `tip-to-tip distance=0.3m`. Moreover, for the state space, all robots move on a 2D plane within `(0,4m)*(0,4m)` (horizontal for Point Robot and Dubins Car but vertical for Planar Quadrotor) with a static rectangle obstacle located at the center coordinate `(2m,1m)` with sizes of `1m*2m`. More specifically, the states of Dubins Car are 2D positions and orientation angle within the 3-dim hyper-rectangle `(0,4)*(0,4)*(0,π)` and the unsafe states are within 3-dim hyper-rectangle `(1.5,2.5)*(0,2)*(0,π)`. The states of Point Robot are 2D positions and 2D velocities within the 4-dim hyper-rectangle `(0,4)*(0,4)*(-1, 1)*(-1, 1)` and the unsafe states are within 4-dim hyper-rectangle `(1.5,2.5)*(0,2)*(-1, 1)*(-1, 1)`. The states of Planar Quadrotor are 2D positions, orientation angle, 2D velocities and angular velocity within the 6-dim hyper-rectangle `(0,4)*(0,4)*(-0.1,0.1)*(-1,1)*(-1, 1)*(-1, 1)` and the unsafe states are within 6-dim hyper-rectangle `(1.5,2.5)*(0,2)*(-0.1,0.1)*(-1,1)*(-1, 1)*(-1, 1)`. For the control inputs, Dubins Car adopts speed and angular speed within the 2D rectangle `(-1,1)*(-1,1)`. Point Robot adopts 2D accelerations as control input within the 2D rectangle `(-1,1)*(-1,1)`. Planar Quadrotor adopts the thrust forces exerted by the two motors as control input within the 2D rectangle `(4,6)*(4,6)` to overcome its gravity and move on the vertical plane.

## B.2 Implementation Details

**Data collection.** As shown in the main text, we adopt supervised learning to train the neural CBFs. The data is collected from random trajectories from the safe state space and control input space through the dynamics. To empirically ensure the forward invariance, we discard the second half states and control inputs to avoid the unsafe region of attraction, and only collect the other states and control inputs with the `safe` labels. To collect `unsafe` data, we collect random states from the unsafe state space with a similar amount of safe data for balance. To be more detailed, the time of random trajectory is 10s with the time step of 0.1s and the states in the last 5s are omitted. Also, the trajectories that are less than 5s (*e.g.* when the robot collides with obstacles or goes beyond the feasible states) are discarded as well. By repeatedly initializing random states and collecting

| Number of grids per dimension | | 10 | 20 | 50 | 100 |
|---|---|---|---|---|---|
| Regular training | Ours w/o BaB | 0.329 | 0.437 | 0.875 | 0.953 |
| | Ours w/ BaB | 0.507 | 0.729 | 0.926 | 0.963 |
| Adversarial training | Ours w/o BaB | 0.247 | 0.592 | 0.837 | 0.910 |
| | Ours w/ BaB | 0.519 | 0.730 | 0.899 | 0.941 |

Table 3: The ablation study of the branch-and-bound (BaB) in our proposed method under different grid densities with $\alpha = 0.5$ for Dubins Car.

trajectories with random control inputs through dynamics, we collect 2.5M total pairs of state and control input with 1.4M safe ones for Point Robot, 4.5M total pairs with 2.5M safe ones for Dubins Car and 1.2M total pairs with 0.7M safe ones for Planar Quadrotor as dataset. Then, we randomly choose 10k data as a validation set and use the rest for model training for each robotic dynamics.

**Model training.** During the model training, we adopt the empirical mean of the positive model predictions as the safe set loss [17] and use the projected gradient descent to find the best-case control input to construct the forward invariance condition loss. To enhance the training efficiency, we only consider the training data along the empirical boundary of $|\phi(\mathbf{x})| < 0.1$ for forward invariance condition loss with $\alpha = 0$. For adversarial training, we adopt project gradient descent over an adjacent cube of each state data to maximize the forward invariance loss, then the gradient descent is based on the worst-case projected states. The size of the adjacent cube in the adversarial training is $1/20$ of each dimension. All the models are trained with Adam for 20 epochs with an initial learning rate of 0.01 and a decay rate of 0.2 every 4 epochs. The neural CBFs for Planar Quadrotor are trained with the weight decay of 0.001.

**Verification procedure.** The first step of verifying neural CBFs is to find the hyper-rectangles to over-approximate the boundary, *i.e.* , find the superset of all the roots $\phi(\mathbf{x}) = 0$. Assuming the hyper-rectangles are small enough such that the CBF is continuous and monotonic for each dimension of the state, we check all the gridded hyper-rectangles to find if all the vertices give positive or negative CBF values. Based on the mean value theorem and the assumption above, the roots exist in the hyper-rectangles whose vertices give CBF values with opposite signs. Once the hyper-rectangles are obtained along the boundary, we verify the forward invariance condition based on off-the-shelf neural network verification toolboxes [56, 57, 58, 59]. More specifically, given the state specification, we first approximate the optimal control input using one vertex and then find the linear bounds based on TaylorModels.jl [43]. Then with the Jacobian bounds [46], the condition in Theorem 3 is found to verify if it is no larger than 0. If it does not hold, we adopt branch-and-bound by half-splitting the state specification along each dimension, and conduct the breath-first search to verify Theorem 3 recursively until reaching maximum iteration of 1000. By maintaining all verified sub-specifications, we approximate the optimal control input for other vertex traversals to verify if the union set of all verified sub-specifications equals the whole state specification. All the baselines are conducted in a similar way but with concretized bounds instead of the symbolic one in Theorem 3. We remark that although the procedure may not be the most efficient due to approximating optimal control input, the verification result is sound and the scalability is satisfactory for current robot dynamics. It is marked as future work to make the procedure more efficient for robot dynamics with higher dimensions.

### B.3 Additional Results and Analysis

**Comparison with different $\alpha$ in the verification condition.** As shown in Figure 5, we compare our results and the branch-and-bound based baseline BBV [36] with different $\alpha$ in the verification condition with different grid numbers per dimension. The upper bound indicates how challenging the verification will be with different $\alpha$. It can be seen that with larger $\alpha$, the performance of BBV decreases more than ours due to larger over-approximation errors. The reason why our results can get

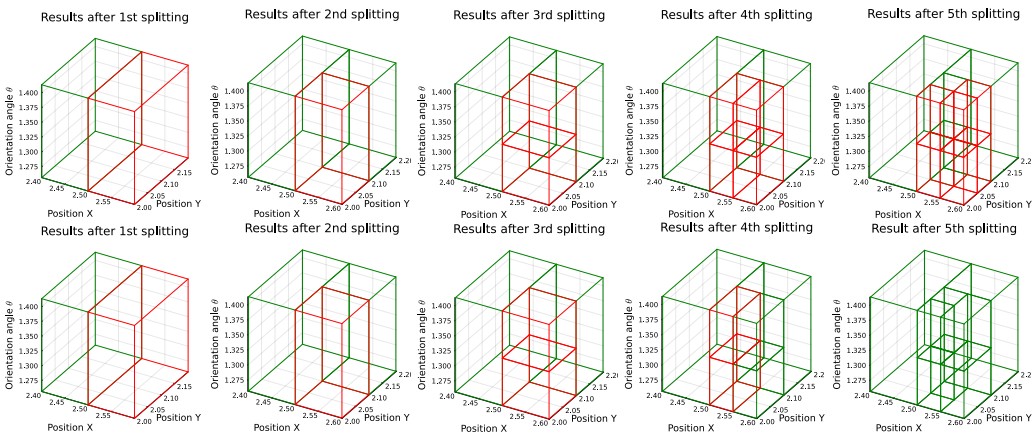

Figure 6: Visualization of branch-and-bound for both baseline BBV [36] (first row) and ours (second row) for Dubins Car with regular-training CBF and $\mathbf{u}_v = [1, 1]$. For each branch after splitting, the green boxes indicate verified specifications while the red ones indicate unknown specifications.

better when $\alpha$ goes up lies in that the approximation errors of gradient bounds through ReLU [46] become less dominant and the symbolic property becomes more significant. Besides, with more fine-grained state specifications, the influence of $\alpha$ becomes less because of fewer approximation errors of both interval arithmetic and ReLU gradient. Adversarial training can also help boost the verification performance of BBV when the grid size is large, while it can hurt our performance with small grid sizes due to more training noise during projected gradient descent.

**Influence of Branch-and-Bound (BaB) on the proposed method.** Here we conduct an experiment as an ablation study to show the influence of branch-and-bound in the proposed method. As shown in Table 3, we can find that without branch-and-bound, the performance is significantly reduced, especially with fewer boundary hyper-rectangles (larger sizes of grids), showing that branch-and-bound scheme is essential to the proposed method to alleviate the extra approximation error caused by finding gradient bounds through ReLU [46].

**Visulizaton of branch-and-bound scheme.** From Figure 6, we can see that after each splitting for the previous unknown specifications, branches will be doubled and the branch-and-bound follows breadth-first search. With fewer split branches, it can be seen that the coarse specifications cannot be verified for both base BBV [36] and ours. However, with more splitting, ours can successfully verify all branches after 5 splits while BBV can only verify some of the finer specifications, leaving lots of unknown branches to be further split due to looser bounds and larger over-approximation. The visualization shows that even though with the same approximated optimal control input $\mathbf{u}_v$, ours can give tighter bounds for neural CBF verification with much fewer split times, resulting in a higher verified rate and shorter verification time.

**Comparison with symbolic interval analysis based method.** Since ReluVal [60] is also based on symbolic bounds and adopts iterative interval refinement, we fairly compare it with CROWN-based baseline BBV [36] and ours with branch-and-bound in Table 4. The experiment is conducted under Dubins Car dynamics with the same amount of maximum split times (1000) of BFS and 10 grids per dimension for input specs. From the result table, we can see that ours still has the best performance. With the default size (16,64,16,1) model, ReluVal performs better than BBV and keeps on par with ours, showing that symbolic ReluVal bounds work well with relatively small model complexity. However, with a larger model size (64,128,64,1) and more unstable ReLU neurons, the bounds of ReluVal become looser, and ReluVal is more likely to perform worse than CROWN-based methods. The main reason lies in different symbolic bounds for crossing-zero unstable ReLU layers. Ours and BBV adopt fully linear symbolic bounds as CROWN does, while ReluVal only adopts symbolic

| Verfication condition | $\alpha = 0.1$ | | $\alpha = 0.5$ | | $\alpha = 1.0$ | |
| Model size | Default | Large | Default | Large | Default | Large |
|---|---|---|---|---|---|---|
| BBV [36] | 0.450 | 0.564 | 0.371 | 0.486 | 0.357 | 0.329 |
| ReluVal [60] | 0.471 | 0.557 | 0.507 | 0.479 | 0.536 | 0.450 |
| Ours | **0.486** | **0.593** | **0.507** | **0.564** | **0.536** | **0.486** |

Table 4: Comparison between CROWN-based baseline BBV [36] and ours with symbolic interval analysis based method ReluVal [60] under different model sizes.

bounds when the lower bound of symbolic pre-activation upper bound is less than 0. Therefore, CROWN-like symbolic bounds are generally tighter than ReluVal [48].

## C    Limitation

One limitation of this work is that it requires an analytical control-affine dynamic model and is not directly applicable to work with (non-control-affine) neural network dynamic models [61, 62, 45]. Besides, it is challenging to efficiently verify neural CBF with high dimensions ($> 10$) of state due to the exponential growth of the number of boundary boxes. Since the scalability also highly depends on the property of the neural CBF, e.g., unstable neuron patterns, local Lipschitiz, etc. Future work can involve robust or certified training techniques [63, 64, 65, 66], to enhance the scalability of verification, e.g., new loss from verification conditions, reducing the number of unstable neurons, etc. Future work could also incorporate the linear bounds into verification-in-the-loop training for verified neural CBFs in real-world applications.

