# OpenReview forum: "Verification of Neural Control Barrier Functions with Symbolic Derivative Bounds Propagation"
_robot-learning.org/CoRL/2024/Conference — CoRL 2024_

### Official Review · Reviewer_mp6J · 2024-07-20

**Originality:** 3
**Technical Quality:** 3
**Clarity Of Presentation:** 5
**Potential Impact:** 3
**Recommendation:** 3
**Confidence:** 3

**Review:**

$\textbf{Quality and Clarity:}$ The paper is well-written and clear to the reviewer, except for a few minor grammar mistakes/typos scattered throughout. The reviewer especially appreciates the concise technical background provided in Section 2. The technical content in Section 3 is also well-presented.

$\textbf{Originality:}$ The original contribution of the paper is the formulation of the symbolic derivative bounds propagation for verifying ReLU-activated MLP-based CBFs for control-affine systems. The contribution is novel as far as the reviewer understands it.

$\textbf{Significance:}$ The proposed approach provides tighter bounds than the existing interval arithmetic based method, resulting in increased verified rates as compared to the baselines. The reviewer believes this is a significant achievement. The authors also claim shorter verification times with their method, although the reviewer is not fully convinced on this point (elaboration in $\textbf{Weaknesses}$).

$\textbf{Strengths:}$ The authors provide a helpful example in Section 3 to clarify their approach. They also show that their bounds formulation contains the interval arithmetic case and thus yields only tighter bounds. This advantage is convincingly demonstrated in their experiments. The parameters along which the experiments vary are well-chosen.

$\textbf{Weaknesses:}$
 - The reviewer is not convinced of the claimed verification time improvement, since it seems to vary depending on the nature of the underlying safety solution, the model size, and the grid size. It would be helpful for the authors to provide an analysis on the conditions that result in better/worse verification time.
 - The proposed method is specific to ReLU-activated MLPs. However, other neural network architectures, e.g., physics-informed neural networks (PINNs) with sinusoidal activations, have also become popular for modeling safety functions and their gradients.
 - The method does not scale well with the state dimension.

**Quality Of The Limitations Section:**

2

**Questions For Rebuttal:**

- The reviewer believes it should be clarified in the abstract that the proposed method is specific to ReLU-activated MLPs.
 - It is not clear how the optimization over the state space in Equation (7) and in Example 1 is achieved.
 - What is the significance of $\alpha$, and how does the choice of $\alpha$ affect the proposed method? Is it fair to use $\alpha \neq 0$ during experiment comparisons?
 - What factors influence the magnitude of the overapproximation errors during the bound computation?
 - It is unclear to the reviewer how the $\textit{approximated}$ control input is computed, e.g., in Example 2.
 - What advantages do CBF-based approaches have over reachability-based approaches (which are guaranteed to yield provably optimal safety functions) for tractable low-dimensional systems, especially since the proposed verification approach scales intractable with the state dimension?

**Robotics Focus:**

3

**Summary Of Paper:**

Efficient and accurate verification of learning-based control barrier functions (CBFs) remains an important and difficult challenge. The authors propose a novel verification method for ReLU-activated multilayer perceptron (MLP) networks based on symbolic derivative bounds propagation. In experiments on different systems and model complexities, the authors demonstrate that their method achieves superior verified rate and verification time as compared to state-of-the-art baselines.

**Summary Of Recommendation:**

The reviewer recommends that the paper is accepted. The paper presents a modest but tangible improvement to existing interval arithmetic based verification methods. The reviewer believes that the proposed symbolic derivative bounds propagation is highly relevant and a good baseline for future work to improve upon.

---

### Official Review · Reviewer_KbLk · 2024-07-21
**This paper is well-written and organized, clearly presenting an efficient verification approach for verifying neural control barrier functions with promising performance demonstrated on various robot systems.**

**Originality:** 4
**Technical Quality:** 5
**Clarity Of Presentation:** 5
**Potential Impact:** 4
**Recommendation:** 4
**Confidence:** 3

**Review:**

This paper is well-written and organized. It clearly presents all relevant information and the reasoning and technical steps behind the development of the verification approach for verifying neural control barrier functions. The efficacy of the proposed approach is validated by extensive experiments.

Strengths:
- Well-written and to the point.
- Mathematically Sound: The basis for the framework (linear bounds for dynamics, bounding Jacobian of neural CBF, bound propagation through inner product) is adequately introduced, laying out a clear foundation for the verification algorithm.
- The numerical examples are constructive for understanding the technical approaches
- Extensive experiments validate the efficacy of the proposed approach

Weaknesses:
- To compute the control signal from the QP program associated with the CBF $\phi(x)$, $\dot \phi(x)$ is required. However, it is unclear how $\dot \phi(x)$ can be computed, considering that $\phi(x)$ uses the ReLU activation function, which is not differentiable.

**Quality Of The Limitations Section:**

3

**Questions For Rebuttal:**

1.	Please explain the meaning of “symbolic”  and  “incomplete verification” somewhere in the manuscript (for readers who are not in the field of verification)
2.	Please explain how to compute $\dot \phi(x)$ (to compute the control signal) when $\phi(x)$ uses a ReLU function.
3.	In the paragraph right below Theorem 2, it mentions “the optimal control input u is approximated…”. Where does the approximation come from?

**Robotics Focus:**

3

**Summary Of Paper:**

This paper addresses the challenges of verifying control barrier functions (CBFs) parameterized by neural networks (neural CBFs) in safety-critical systems and robot control applications. The authors introduce a novel verification framework that leverages symbolic derivative bound propagation. This method combines the properties of linearly bounded nonlinear dynamic systems with the gradient bounds of neural CBFs, utilizing the Heaviside step function for the derivatives of activation functions. The proposed approach propagates symbolic bounds through the inner product of the neural CBF Jacobian and nonlinear system dynamics. Extensive experiments on various robot dynamics demonstrate that this method outperforms interval arithmetic-based baselines in terms of verified rate and verification time, showcasing its effectiveness and efficiency across different model complexities.

**Summary Of Recommendation:**

The paper is well-written and organized. The symbolic bound propagation based verification approach seems very promising due to the tightness of the bounds achieved through clearly explained steps.

---

### Official Review · Reviewer_sbF7 · 2024-07-24
**The proposed method is incremental and scales poorly with the dimension of the state space limiting its applicability to actual robotic systems.**

**Originality:** 2
**Technical Quality:** 2
**Clarity Of Presentation:** 3
**Potential Impact:** 2
**Recommendation:** 2
**Confidence:** 4

**Review:**

Clarity:
--------
The writing style of the paper needs to be improved. Several sentences are incredibly long, which makes it hard to follow.

Originality:
------------
The paper uses symbolic interval analysis to propagate bounds over the Jacobian of the neural network. Symbolic Interval Analysis has been heavily studied for formal verification of NNs. See for example:

[*] Wang, Shiqi, Kexin Pei, Justin Whitehouse, Junfeng Yang, and Suman Jana. "Formal security analysis of neural networks using symbolic intervals." In 27th USENIX Security Symposium (USENIX Security 18), pp. 1599-1614. 2018.
This paper uses a similar concept but on the Jacobian of the NN instead of the NN itself.

Strengths:
------------
- The evaluation section shows the proposed method achieves better results than other methods.

Weaknesses:
--------------
- A major weakness of the proposed method is the need to partition the space of states and control actions, limiting the proposed framework's scalability.

- The paper claims that one of the main contributions is to solve the "quadratic operator of the inner product." But this problem has never been clearly defined. Each section forward points to the next one, but a clear definition of the problem is never well articulated.

Some technical questions need to be addressed. The "Questions For Rebuttal" below details these questions.

**Quality Of The Limitations Section:**

2

**Questions For Rebuttal:**

(1) What is the difference between $\phi$ and $\phi_\theta$ (it seems the paper is using interchangeably without any mention of that)?

(2) In Section 2.2, the paper states that one of its objectives is to ensure that the sublevel set of $\phi$ is a valid subset of the user-specific safe set. Later in the same paragraph, the paper abandons this objective by assuming that one can modify the pre-trained NN (by adding a constant C) to achieve such a goal. The paper should explain how to find such a constant C since it affects the overall time for verifying the CBF. Was the time taken to find C accounted for in the experiments?

(3) In Section 3.2, the paper states that "Equation (7) avoids intractable linear bound propagation through the quadratic operator of the inner product." It is not clear why linear bound propagation is intractable. Off-the-shelf tools are capable of propagating linear bounds over a whole set of nonlinearities in polynomial time. A proof showing that linear bound propagation is intractable is needed.

(4) Just after Theorem (2), the paper compares the obtained bound to "concretized and non-symbolic." But what about "symbolic interval analysis" like those used in [*] above?

**Robotics Focus:**

2

**Summary Of Paper:**

Given a pre-trained neural network, the paper proposes a symbolic bound propagation method to verify that the neural network is a Control Barrier Function. The paper proposes

**Summary Of Recommendation:**

The proposed method is incremental and scales poorly to robots with high-dimensional state spaces.

---

### Author Rebuttal · Authors · 2024-08-07

Dear Reviewers and ACs/PCs,

We thank all the reviewers and AC for their time and valuable suggestions.

Following the suggestions from the reviewers, we have conducted additional experiments and more clarification and discussion in our revision. More specifically, we added the definition of bound propagation through the inner product and highlighted the focus of ReLU-based MLPs. We also conducted extra experiments to compare the symbolic interval analysis based method and ours. Besides, we clarified the time comparison with baselines under different settings network size and grid size and added more discussion to enhance the scalability of the proposed method.

The revised PDF has been zipped and uploaded to OpenReview, with major revisions highlighted in blue. We would be happy to discuss if there are any other concerns about our work. Thanks again for the suggestions and comments.

Best,

Authors of submission 253

---

### Decision · Program_Chairs · 2024-09-04

**Decision:**

Accept

**Comment:**

Strengths: Overall the paper is well-written (but some very long sentences could do with rewrites). The symbolic derivative bounds propagation for verifying ReLU-activated MLP-based CBFs for control-affine systems is a novel contribution. Tighter bounds than the existing interval arithmetic-based method, resulting in increased verified rates as compared to the baselines.

Weaknesses: There is some concern about the scalability of the method and the claimed verification time improvement. The method also appears to be specific to ReLU-activated MLPs. The reviews provide detailed questions for rebuttal.

----------

As part of the rebuttal, I'd like to commend the authors on clarifying their contributions. I think the strengths of the paper outweigh its weaknesses.